# Herbal Products Used in Menopause and for Gynecological Disorders

**DOI:** 10.3390/molecules26247421

**Published:** 2021-12-08

**Authors:** Maša Kenda, Nina Kočevar Glavač, Milan Nagy, Marija Sollner Dolenc

**Affiliations:** 1University of Ljubljana, Faculty of Pharmacy, Aškerčeva Cesta 7, 1000 Ljubljana, Slovenia; masa.kenda@ffa.uni-lj.si (M.K.); nina.kocevar.glavac@ffa.uni-lj.si (N.K.G.); 2Comenius University in Bratislava, Faculty of Pharmacy, 83232 Bratislava, Slovakia; nagy@fpharm.uniba.sk

**Keywords:** menopause, dysmenorrhea, premenstrual syndrome, gynecological disorders, herbal products, medicinal plants

## Abstract

Herbal products are often used as an alternative to pharmacological therapy. Menopausal symptoms and gynecological disorders (such as premenstrual syndrome and dysmenorrhea) are the indications where pharmacological therapy may have serious adverse events, hence many women prefer to use herbal products to help with these symptoms. Here, we reviewed plants and derived products, which are commonly used for the abovementioned indications, focusing on clinical data, safely profile and whether or not their use is justified. We noted that limited data are available on the use of some plants for alleviating the symptoms of menopause and gynecological disorders. While black cohosh (*Cimicifuga racemose*) and red clover (*Trifolium pretense*) were consistently shown to help reduce menopausal symptoms in clinical studies, currently available data do not fully support the use of fenugreek (*Trigonella foenum-graecum*), hops (*Humulus lupulus*), valerian (*Valeriana officinalis*), and soybean (*Glycine max* and *Glycine soja*) for this indication. For premenstrual syndrome and premenstrual dysphoric disorder, chaste tree (*Vitex agnus-castus*) shows effectiveness, but more clinical studies are needed to confirm such effect upon the use of evening primrose (*Oenothera biennis*).

## 1. Introduction

Women often seek help for various gynecological disorders. Most commonly, these are premenstrual syndrome, dysmenorrhea, and menopausal symptoms. Often, they prefer alleviation of symptoms with herbal products over pharmacological therapy [1]. This is especially the case with, e.g., hormone replacement therapy in menopause, as this therapy bears the possibility for serious adverse events, such as breast cancer [2].

Premenstrual syndrome is characterized by irritability, tension, depressed mood, breast tenderness and bloating in the weeks before menstruation [3]. These symptoms are severe in 5–8% of women. Typical pharmacological therapies include analogues of gonadotropin-releasing hormone, estradiol, contraceptives and serotonin reuptake inhibitors.

Dysmenorrhea is the occurrence of painful cramps of the uterus [4]. Here, we will focus on primary dysmenorrhea, which is present due to menstruation as opposed to secondary dysmenorrhea, which can have different underlying reasons, such as endometriosis, pelvic inflammatory disease, ovarian cysts, and adenomyosis, to name a few [5]. Typical therapies for primary dysmenorrhea are contraceptives, progestins and non-steroidal anti-inflammatory drugs [4].

Menopause occurs approximately one year after the last menstruation cycle, which stops due to the gradual decrease in ovarian function [6]. The mean age of women entering menopause is 51 years. The transition period, characterized by the cessation of ovarian function, is called perimenopause and starts several years prior to menopause. Women entering perimenopause and menopause experience several symptoms, which are assessed by different criteria. Kupperman menopausal index is often used to measure the intensity of menopausal symptoms (hot flashes, excessive sweating, sleep disturbances, irritability, depressive mood, attention deficit disorder, joint and bone pain, headache, arrhythmias, paresthesia) assessed on a 1 through 4 scale [7]. Other similar assessments are done with the Greene climacteric scale, or the menopause rating scale [8]. Pharmacological treatments for these symptoms include hormone replacement therapy, selective serotonin reuptake inhibitors and selective serotonin-norepinephrine reuptake inhibitors [6,9,10,11]. Often, the effectiveness of therapy is measured by evaluating the frequency and severity of hot flashes. The symptoms of menopause tend to fade away over time—generally in a year. This makes trials of supplements intended to relieve these symptoms very intricate and exemplifies the need for a control group.

Here, we reviewed plants and derived products, which are commonly used in the Western world as dietary supplements or over-the-counter drugs for the abovementioned indications (the use of some is supported by the European Medicines Agency herbal monograph), focusing on clinical data, safely profile and whether or not their use is justified.

## 2. Results

### 2.1. Black Cohosh (Actaea racemosa L./Cimicifuga racemosa (L.) Nutt.)

Black cohosh or *Cimicifuga racemosa* syn. *Actaea racemosa* is a perennial and is endemic to the eastern United States and Canada. This plant belongs to the family Ranunculaceae. It forms up to 2 m of creeping rhizomes. It has elongated fringed divided leaves [12]. Inflorescences of small, white flowers in the form of long clusters appear at the end of branched stems from May to August. The rhizome is used as a herbal drug for medicinal purposes. Native American tribes living in the area of growth of this plant have traditionally used it for centuries [12,13]. Common names for this herb are black cohosh, macrotys, rattleweed and black snake root. In some European countries, the herbal preparations of black cohosh rhizome are marketed as herbal medicines with proven use to relieve menopausal symptoms, e.g., hot flashes. In the United Kingdom, black cohosh is a traditional drug for the symptomatic relief of rheumatic pain [13,14].

There are three groups of compounds in black cohosh that are responsible for its pharmacological action: phenolic compounds, including ferulic acid, isoferulic acid and caffeic acid derivatives, cycloartane triterpene glycosides (actein, 26-deoxyactein) and phenylpropanoids. The phytoestrogenic flavonoid formononetin was also found in early studies of the constituents of this plant [15], but this was not confirmed in neither the raw herb nor the standardized extracts in later studies [12]. Despite numerous research efforts, we do not know the exact composition or function of this plant. The preparations of black cohosh are standardized to the triterpene glycoside content of 26-deoxyactein. Other important triterpene glycosides are actein and cimicifugoside (aglycone cimegenol) [12].

A black cohosh extract contains many triterpene glycosides [16], but there is no direct evidence that these are the main active ingredients in relieving menopausal symptoms, especially hot flashes [13]. The efficacy of black cohosh extract in reducing hot flashes may be attributed to the binding and modulation of key central nervous system receptors for thermoregulation, mood, and sleep (e.g., receptors for serotonin, dopamine, γ-aminobutyric acid (GABA), µ-opioids). It also affects the improvement of metabolism in the brain and its overall activity [17,18]. The extract contains active ingredients that act as partial agonists of the serotonin receptors (also known as 5-HT or 5-hydroxytryptamine receptors), which are located in the hypothalamus and are associated with thermoregulation. From the 75% ethanol extract standardized to 5.6% triterpene glycosides, the compound *N_ω_*-methylserotonin was isolated, which could be the main active ingredient of the extract [19]. Another mode of action of the rhizome extract could be via triterpenoids (especially 23-*O*-acetylshengmanol-3-*O*-d-xylopyranoside) through modulation of GABA_A_ receptors [20]. The triterpenoid deoxyactein has also been associated with beneficial effects in osteoporosis by influencing osteoclast growth and differentiation and mineralization [21,22].

Castelo-Branco and coworkers, in a systematic review of the literature, found 35 clinical trials and one meta-analysis involving 43,759 women, 13,096 of whom were treated with an isopropanol extract of black cohosh [23]. Based on these data, it was found that both neurovegetative and psychological menopausal symptoms were significantly less pronounced when the extract was used than in the group of women who did not receive it. Higher doses have also been found to be more effective, especially when combined with St. John’s wort (*Hypericum perforatum*) [24]. The effect of the isopropanol extract can be compared even with low doses of transdermal estradiol or tibolone and has a better benefit-risk profile than tibolone [25]. Few side effects occurred, which were not more severe than in the placebo group, and no hepatotoxic effects were observed [23].

Based on these results, it can be concluded that the benefits of using black cohosh extract or a combination of black cohosh extract and St. John’s wort outweigh the risks, so that their use is recommended for the treatment of menopausal symptoms [23], including in patients with hormone-dependent disorders [26,27].

### 2.2. Chaste Tree (Vitex agnus-castus L.)

Chaste tree or *Vitex agnus-castus* L. is a deciduous shrub or a small tree of the Lamiaceae family, native to areas stretching from the Mediterranean to northern India. It grows to a height of 6 m. It has pale violet panicular inflorescences that ripen into brown fruits with a characteristic, aromatic and peppery aroma [28,29]. The fruits have been used traditionally for their emmenagogue, lactagogue, vulnerary, carminative, antihelmintic and anti-inflammatory properties [30], while in terms of rational phytotherapy, they are considered herbal substances most frequently used in the treatment of premenstrual syndrome [29]. *V. agnus-castus* is also believed to have been used by monks as an anaphrodisiac, to diminish their sex drive, hence the common names monk’s pepper, chaste tree and chasteberry [29].

Phytochemical compounds of the fruit include volatile compounds (essential oil), flavonoids and other phenolic compounds, iridoids, ketosteroids and diterpenoids [30,31]. In 2016, an LC/MS method for differentiation between the chaste tree and two related species popular in Japan, *V. rotundifolia* and *V. trifolia*, was described, based on the identification of chastol and epichastol diterpenoids proposed to be marker compounds in chaste tree fruits [32,33].

According to the European Pharmacopeia [34], whole, ripe and dried fruits are used as a herbal substance (*Agni casti fructus*). Casticin, also known as vitexicarpin, which is a methoxylated flavonol, is defined to a minimum of 0.08% content in the dried herbal substance. In addition, the extract of chaste trees (*Agni casti fructus extractum siccum*) is prepared from the herbal substance by a suitable procedure using ethanol (40–80%, *v*/*v*) and must contain a minimum of 0.1% casticin (dried herbal substance) [34].

The largest body of evidence focuses on the use of chaste trees in the treatment of premenstrual syndrome. The Committee on Herbal Medicinal Products (HMPC) of the European Medicines Agency (EMA) concluded that a dry extract (DER (drug-extract ratio) 6–12:1; ethanol 60%, *m*/*m*) can be used over three months, 20 mg once daily, for the treatment of premenstrual syndrome (a well-established herbal medicinal product), while on the basis of long-standing use other preparations can be used for the relief of minor symptoms of the premenstrual syndrome (a traditional herbal medicinal product) [35].

The main mechanism of action is believed to be dopaminergic action, i.e., binding to dopamine receptors followed by a decreased release of prolactin. Rotundifuran, 6β,7β-diacetoxy-13-hydroxy-labda-8,14-diene and clerodadienols were shown to play a crucial role [36,37]. The involvement of the serotoninergic system has also been proposed, particularly in connection with premenstrual dysphoric disorder [38]. The effectiveness of chaste trees has been continuously researched since the first clinical studies in the 1990s. The evaluation of clinical results based on questionaries’ assessments was conducted in references [39,40,41,42,43,44,45,46,47,48,49,50], while pharmacological parameters such as serum prolactin measurements were observed in references [29,51,52]. Studies are described briefly in the following.

A double-blind, randomized, placebo-controlled clinical study from 1993 [39] focused on the relief of symptoms of premenstrual syndrome, using Moos’ menstrual distress questionnaire. Significant improvement was shown for the ‘feel jittery or restless’ symptom, while no differences between the verum and placebo groups were found for impaired concentration, fluid retention and pain. In 1997, premenstrual tension syndrome was evaluated in a randomized, controlled study vs. pyridoxine, involving 175 women [40]. Effectiveness was assessed using the premenstrual tension syndrome scale, the recording of six characteristic complaints of the syndrome and the clinical global impression scale. In the chaste tree group, there was a significant reduction in breast tenderness, edema, inner tension, headache, constipation and depression. In a prospective, multi-center study [41], 43 patients were observed during eight menstrual cycles; 13 were receiving concomitantly oral contraceptives. According to the results of the Moos’ menstrual distress questionnaire, a visual analogue scale and a global impression scale, premenstrual syndrome was successfully reduced, and symptoms were evaluated in connection to the luteal/follicular phase. No differences were seen between patients on or off oral contraceptives. Schellenberg et al. [42] designed a randomized, double-blind, placebo-controlled, parallel-group comparison study involving 170 women with premenstrual syndrome, over three menstrual cycles. Women self-assessed irritability, mood alteration, anger, headache, breast fullness and bloating, and improvement was statistically greater in the verum group. The clinical global impression scale index also improved significantly, and responder rates were 52% for the verum group and 24% for the placebo. Cyclical mastalgia was the focus of a double-blind, randomized, placebo-controlled study on 97 women over three menstrual cycles [43]. The intensity of mastalgia was recorded once per cycle using a visual analogue scale. The differences were significantly greater between the verum and the placebo group. Most of the questionnaire-based studies conducted in the 2000s also confirm the use of chaste trees as an effective treatment for premenstrual syndrome. In the group of 67 women (a prospective, double-blind, randomized, placebo-controlled study; [44]) and 208 women (a prospective, double-blind, placebo-controlled, parallel-group, multi-center study; [45]), respectively, significant improvements were found based on patients’ assessments in premenstrual syndrome diary or the premenstrual tension syndrome self-rating scale. In another double-blind, randomized, placebo-controlled study on 128 women, headache, anger, irritability, depression, breast fullness, and bloating and tympani symptoms were shown to significantly improve after six menstrual cycles [46]. A significant decrease of severity of total symptoms of the premenstrual syndrome, as well as improvement in psychological and physical symptoms, were observed in a randomized, double-blind, placebo-controlled, parallel-group comparison over two menstrual cycles on students [53].

Premenstrual dysphoric disorder was the focus of a randomized, single-blind study controlled with fluoxetine, a serotonin reuptake inhibitor [48]. The Penn daily symptom report, the Hamilton depression rating scale, and the clinical global impression scale-severity of illness and improvement were used for the evaluation of results after two months of treatment. Fluoxetine was more effective in terms of psychological symptoms and the chaste tree extract diminished physical symptoms, while there was no statistically significant difference with respect to the rate of responders in both groups. Another fluoxetine-controlled, double-blind, randomized study concluded that chaste tree offers an effective treatment of premenstrual dysphoric disorder, however, it was outperformed by fluoxetine on all endpoints [49].

Three randomized studies included women suffering from hyperprolactinaemia. A double-blind study vs. placebo on 52 women [51] studied a daily dose of 20 mg of chaste tree preparation taken for three months, and 37 complete reports were evaluated. Significance in the reduction of prolactin release, and normalization of luteal phases and progesterone synthesis was observed. A prospective study involved 40 women with cyclic mastalgia and 40 women with hyperprolactinemia who received a three-month treatment of either bromocriptine or chaste tree extract [52]. Pre- and post-treatment serum prolactin and breast pain were evaluated, and significantly lower levels of prolactin and mastalgia pain were observed in both groups, while there were no significant differences between the groups. In a pilot study by Wuttke et al. [29], 56 women with mastalgia, who received a chaste tree combination product for the period of three menstrual cycles, experienced a significant reduction of serum prolactin levels as compared to a placebo.

Not directly related to premenstrual syndrome, but showing the hormonal activity, is a prospective, double-blind, randomized, placebo-controlled study of Gerhard et al. [54] who investigated chaste tree effects in 96 women with fertility disorders (secondary amenorrhoea, luteal insufficiency, idiopathic infertility). After three months of therapy, pregnancy or spontaneous menstruation was achieved significantly more often in the verum than the placebo group.

A dose-effect relationship for the treatment of premenstrual syndrome was studied in 2012 [50]. The chaste tree extract with 60% ethanol (*m*/*m*) and a DER 6-12:1, standardized to casticin, was used in doses of 8, 20 and 30 mg over three menstrual cycles. The multicenter, randomized, double-blind, placebo-controlled, parallel-group study in 162 women evaluated the symptoms of irritability, mood alteration, anger, headache, bloating and breast fullness. Improvement in the total symptom score was significantly higher in the 20 mg group than in the placebo and 8 mg treatment group, while the higher dose of 30 mg showed no additional improvements.

Systematic analyses of clinical studies are reviewed in [55,56,57]. Based on the reviewed data about randomized controlled studies, authors of [55,56] conclude that chaste tree is effective and safe in the treatment of premenstrual syndrome and premenstrual dysphoric disorder. The methodological quality of studies was generally moderate to high. Main limitations include, for example, different chaste tree preparations, different diagnostic criteria and small sample sizes. However, authors in [57] have concluded that the real treatment effect may be overestimated due to a high risk of bias, high heterogeneity and risk of publication bias.

Side effects, if observed, were mild and less severe than in the control treatments. The main symptoms reported were headaches, skin reactions, acne, urticaria and gastrointestinal complaints [55].

### 2.3. Evening Primrose (Oenothera biennis L.)

Evening primrose or *Oenothera biennis* L. is a biennial plant that grows to between 30 and 150 cm in height. It is native to Central America, from where it spread first to North America and Europe. It grows today in regions with a moderate climate all over the world. The plant was named after the pleasant spectacle that unfolds every evening at dusk, when its yellow flowers open. However, they only last until the following day. Fruits are up to 4 cm long and contain numerous small seeds [58,59]. The plant’s stem and leaf juices and poultices were used by Native Americans dermally to treat skin inflammation, bruises and minor wounds, while the leaves were used internally for gastrointestinal disorders and sore throats [60]. Today, the seed oil is well-known, particularly in alternative treatments, for use in inflammatory conditions such as atopic dermatitis, eczema and rheumatoid arthritis, and women’s conditions, which are described in the following sections.

The seeds of evening primrose contain approx. 20% of oil (triglycerides), which is yellow to greenish-yellow when unrefined, with a typical odor. Linoleic acid is a highly predominant fatty acid (typically ~70%) followed by γ-linolenic acid (typically ~10%) [58]. Both belong to the group of polyunsaturated omega-6 fatty acids. The European Pharmacopeia specifies limits for individual fatty acids, i.e., palmitic (4–10%), stearic (1–4%), oleic (5–12%), linoleic (65–85%), γ-linolenic (7–14%) and α-linolenic (max 0.5%) acids, as well as unsaponifiable matter (max. 2.5% determined on 5 g of oil) [34]. Due to the high oxidative instability of the oil, it is important that manufacturers ensure the peroxide value of max. 10.0 (or max. 5.0 if intended for use in parenteral preparations [34].

Regulatory accepted in the European Union is the use of oil obtained from two *Oenothera* species, *O. biennis* L. and *O. lamarckiana* L., in the form of a traditional herbal medicinal product for the symptomatic relief of itching in acute and chronic dry skin conditions, exclusively based upon long-standing use [61].

The mechanism of action of *Oenothera* oil is attributed to the effects of omega-6 fatty acids on immune cells and the synthesis of prostaglandins, cytokines and cytokine mediators [60]. It is also assumed that low levels of prostaglandin E1 in women with premenstrual syndrome lead to increased sensitivity to luteal phase prolactin [62]. Research studies and findings which proposed a possible connection between premenstrual syndrome, prolactin levels, prostaglandins and γ-linolenic acid, an essential fatty acid precursor of prostaglandin E1, originate from the early 1980s [63,64], including first clinical studies showing success in the treatment with *Oenothera* oil, for premenstrual syndrome [65] and mastalgia [66]. Data showed that women with premenstrual syndrome have the inability to convert linoleic acid to γ-linolenic acid due to a decreased activity of the delta-6 desaturase enzyme [67], in addition to hormonal disbalance [62]. Cerin et al. [62] investigated hormonal status (progesterone, estradiol, prolactin, cortisol, aldosterone) and cholesterol, triglyceride, lipoprotein, magnesium and calcium levels, and glucose tolerance in the follicular and luteal phases of the menstrual cycle in women diagnosed with premenstrual syndrome vs. symptom-free controls. The parameters were found to be similar in both groups, except for aldosterone which was lower in the follicular and luteal phases, and cholesterol which was higher in the follicular phase in women with premenstrual syndrome. In the same study, the effect of *Oenothera* oil was also evaluated in a randomized, double-blind crossover design, but no effects were found for any of the biochemical parameters.

Questionnaire-based clinical studies also show contradictory results. Studies by Pruthi et al. [68], Pye et al. [66], Mcfayden et al. [69] and Blommers et al. [70] reported improvements of mastalgia symptoms, although some results were not significant, Kashani et al. [71] reported improvement in overall premenstrual syndrome, and Kazemi et al. [72], Mehrpooya et al. [73] and Cancelo Hidalgo et al. [74] reported a reduction of climacteric symptoms; the latter study used a combination of *Oenothera* oil, isoflavones and vitamin E. On the other hand, the effectiveness in relieving symptoms of premenstrual syndrome was not confirmed [75,76,77,78], nor was the effectiveness in relieving menopausal flashing [79]. Benefits in the treatment of premenstrual syndrome seem to be at a level of a placebo effect.

Doses of *Oenothera* oil used in the studies were between 500 and 3000 mg/day, alone or in combination with vitamin B6 or vitamin E, while the largest dose of 6000 mg/day was used in [76].

Systematic literature and study reviews were done by Budeiri et al. [80], Low Dog [81], Dante et al. [82], Srivastava et al. [83], Stevinson et al. [84] and Cheema et al. [10], and *Oenothera* oil was not found to be significantly better than a placebo. However, since approaches of complementary and alternative medicine are widely used by women with premenstrual syndrome and menopausal symptoms, including *Oenothera* oil [85,86,87], and since *Oenothera* oil treatments were not found to be associated with major side effects, health care practitioners are encouraged to conduct informed discussions with patients and recognize situations where it may be reasonable to recommend the *Oenothera* oil therapy, together with suitable nutritional approaches, the use of exercise and mind–body techniques [67,88].

### 2.4. Fenugreek (Trigonella foenum-graecum L.)

Fenugreek or *Trigonella foenum-graecum* L. is a member of the Fabaceae family. It is grown in the Mediterranean, northern Africa and Indian peninsula to be used as a herb, spice, or in traditional medicine [89,90]. Fenugreek is an annual plant, which grows up to the height of 60 cm, has trifoliate leaves and white to yellow flowers [90]. Fenugreek seeds grow in thin pods, which are about 15 cm long, and are a part of plant which is commonly used in formulations, such as powder, dry extract or soft extract [89]. The seeds are golden yellow and contain polysaccharides (24–25% galactomannans), 0.016% essential oil, 0.6–1.7% saponins (from diosgenin, yamogenin, tigogenin, and others), sterols (β-sitosterol), flavonoids (orientin, isoorientin, isovitexin) and other secondary metabolites (protoalkaloids, trigonelline, choline) [89]. A wide array of versatile compounds contained in fenugreek seeds is the reason that many health effects have been attributed to this plant. These include antidiabetic, antihyperlipidemic, antiobesity, anticancer, anti-inflammatory, antioxidant, antifungal, antibacterial, galactagogue activities, and to help with climacteric and period problems [90]. In this review, we concentrate on the alleviation of menopausal symptoms and dysmenorrhea.

Diosgenin and yamogenin are steroidal sapogenins, which are obtained after the acid hydrolysis of fenugreek seeds. They are of interest for the pharmaceutical industry due to the possibility to synthesize oral contraceptives and steroid hormone drugs from them, and due to their own pharmacological activity [90]. Another compound with the activity on the endocrine system is the alkaloid trigonelline, which is a phytoestrogen, as it activates the estrogen receptor (ER) [90]. Another set of compounds, which might play a role in alleviating menopausal symptoms and are contained in fenugreek, are flavonoids [90]. Not only trigonelline, but also fenugreek extract showed the ability to bind to ER in a competitive ER binding assay and to have agonist activity on this receptor in a transactivation gene reporter assay [90,91]. Furthermore, fenugreek extract upregulated the expression of the estrogen-responsive gene and induced proliferation of estrogen-dependent breast cell line MCF-7 [91]. Another study showed the antiproliferative effect of fenugreek extract on several breast cancer cell lines, including MCF-7 cells [90,92]. The same cell line was used for subcutaneous implantation in female mice where diosgenin (which is present in fenugreek seed) inhibited tumor growth [90,93]. Extract preparation and active compound content may thus be the underlying reason for different results in these studies.

Fenugreek extract was shown to improve sexual function in women in a randomized placebo-controlled study, where increased plasma 17β-estradiol was measured, possibly due to increased aromatase conversion of testosterone to 17β-estradiol [90,94]. In a placebo-controlled study on 101 women, fenugreek extract helped with dysmenorrhea symptoms during menstruation [90]. Duration of pain was decreased and reduction of systemic symptoms, such as fatigue, headache and nausea, was observed [95]. A study from 2006, where postmenopausal women were given 6 g of fenugreek seed powder for eight weeks revealed improvement of hot flashes and night sweats after four weeks [96]. However, fenugreek seed extract was less effective than hormone replacement therapy in alleviating these symptoms. A recent (2020) randomized, double-blind, placebo-controlled study on perimenopausal women taking 500 mg fenugreek extract for 42 days also showed a more than 20% reduction in hot flashes, night sweats, insomnia, and more than 30% improvement of depression [97,98]. Increases in serum 17β-estradiol, free testosterone and progesterone were observed, and decreases in follicle-stimulating hormone and steroid hormone-binding globulin. Authors speculated that this indicates the phytoestrogenic effect of fenugreek and an establishment of hormonal balance in postmenopausal women upon taking fenugreek extract [98].

In terms of safety and possible interactions, fenugreek is generally safe, but patients taking antidiabetic drugs should monitor their blood sugar regularly [90]. Fenugreek can cause digestive disorders and allergic reactions [89]. It should not be taken during pregnancy due to uterine stimulatory and abortifacient activities [89]. Moreover, non-clinical data suggest embryo-lethal effects, testicular toxicity and decreased thyroid hormone levels [89].

Data on fenugreek usage with the intention to help with menopausal symptoms and dysmenorrhea are too sparse to support its use for these indications. More studies need to be conducted to fully elucidate the potential fenugreek might have in this therapeutic area.

### 2.5. Hops (Humulus lupulus L.)

There are three main species in the genus *Humulus* (Cannabaceae): *Humulus lupulus* L., *H*. *scandens* (Lourr.) Merr. and *H*. *yunnanensis* Hu. [99]. *Humulus lupulus* L. is native to central Europe; however, today it is naturalized throughout the northern temperate regions. It is a perennial and dioecious climbing plant with a herbaceous stem which can reach 10 m. Male small flowers are organized in clusters. The female inflorescences are cones that contain foliaceous bracts and so-called glandular trichomes in the lupulin glands containing essential oil (constituents: β-myrcene, β-caryophyllene, α-humulene, β-farnesene, α-selinene, β-selinene, humulene epoxides, β-bisabolol, 2-methyl-3-buten-2-ol, a.s.o.), prenylated acylphloroglucinols (α-acids: humulone = HU, its derivatives, and β-acids: lupulones), prenylated flavanones (isoxanthohumol = IX, 6-prenylnaringenin = 6PN, 8-prenylnaringenin = 8PN), chalcones (xanthohumol = XH, desmethylxanthohumol), triterpenes, flavonols, and tannins [100]. XH is the main chalcone in the lupulin glands (0.1–1% of cone dry weight) [101]. Although IX, 6PN and 8PN are present in different varieties of hops [102], some authors theorized these derivatives could be partly decomposition products emerging during drying and storage [103]. Female inflorescence of hop is important for the production of beer. Moreover, there is a pharmacopoeial drug monograph [104] used for the quality check during the production of herbal teas or herbal preparations (comminuted or powdered herbal drug; liquid extract (DER 1:1), extracted with ethanol 45% *v*/*v*; liquid extract (DER 1:10), extracted with sweet wine; tincture (ratio of herbal substance to extraction solvent 1:5), extracted with ethanol 60% *v*/*v*; and dry extract (DER 4-5:1), extracted with methanol 50% *v*/*v*). They are all mentioned in a category of traditional herbal medicinal products, used for the relief of mild symptoms of mental stress and to aid sleep [104].

Increased interest in therapeutic use of hops dates back to the end of the last century when it was discovered that hops contains prenylflavonoids which are thought to be phytoestrogens. Milligan and colleagues isolated estrogenic 8PN by bioassay-guided fractionation of hops extracts [105]. It has long been traditionally believed that hops has strong estrogenic activity, e.g., in women, harvesting hops by hand, who started menstruating two days after the hops harvesting began. Koch and Heim claimed that the estrogenic activity of hops corresponds to the presence of the equivalent of 20–300 g 17β-estradiol/g [106]. Since then, more systematic research has begun. The main focus was on the study of the effects influencing the symptoms of menopause.

A prospective, randomized, double-blind, placebo-controlled trial (12 weeks, 67 postmenopausal women) with a hop extract standardized to 8PN was evaluated using responses to a modified Kupperman menopausal index [107]. All groups showed a significant reduction in Kupperman menopausal index at both week 6 and week 12. Hop extract at 100 g 8PN was significantly better than the placebo at 6 weeks but not at 12 weeks. A dose-response relationship could not be established as the higher dose (250 g) was less effective than the lower dose at both 6 weeks and 12 weeks. The unclear dose-response relationship may have been due to the small size of the study group of women. Furthermore, in another study by Erkkola et al. [108] enrolled a small number of women 36. It was a 16-week randomized, double-blind, placebo-controlled, crossover study, with 8 weeks on hops (hydroalcoholic extract containing 0.13% 8PN, 0.12% 6PN, 1.6% IX and 2.72% XH; capsules with 75 mg of extract), then 8 weeks in the opposite treatment arm. Outcome measures included the Kupperman menopausal index, the menopause rating scale, and a multifactorial visual analog scale at baseline and after 8 and 16 weeks. There was no significant change in symptoms in those women who did the active treatment followed by a placebo. These results are not meaningful due to a small number of participants and a large number of variables evaluated. A daily dose of 500 mg hops in tablet form compared with a placebo (90 days, 120 postmenopausal women, 40–60 years, and a minimum 12 months and maximum 5 years after the last menstrual bleeding, or premenopausal women with less than 12 periods during the last 12 months) showed a statistically significant reduction in menopausal symptoms and a strong reduction in hot flashes [109]. This trial assessed the effectiveness of hops on depression and anxiety, too. The depression score showed a statistically more significant decrease in the hops group at the 4th, 8th and 12th weeks compared to the placebo. No side effects were observed in the groups due to intervention.

In a crossover, double-blind, placebo-controlled study, 11 men and 11 women aged 19–37 years consumed 1 L of a drink containing 12 mg/L XH for 14 days [110]. They found no differences in plasma levels of 17β-estradiol, progesterone, osteocalcin, and alkaline phosphatase at the beginning compared with the end of the study. In vivo, it has been confirmed [111] that about 40% of humans have an intestinal microbiota capable of converting non-estrogenic XH to estrogenic 8 PN. In vitro experiments showed that activation of IX to 8PN occurs only in the distal colon. In a preliminary in vivo experiment, the importance of the gut microbiota for 8PN exposure in vivo was demonstrated. One individual from each group with high, medium and low conversion of IX was administered IX for 4 days. The amount of 8PN excretion correlated significantly with in vitro data from stool samples. Legettes and colleagues [112] investigated the pharmacokinetics of a single dose of 20, 60, or 180 mg of pure XH in men (*n* = 24) and women (*n* = 24) using LC-MS/MS. IX conjugates were dominant in all subjects. Levels of 6PN and 8PN were undetectable in most subjects. The pharmacokinetic profile of XH showed maximum concentrations around 1 h and between 4 and 5 h after ingestion. Van Breemen et al. [113] investigated the pharmacokinetics of XH, IX, 6PN and 8PN in a sample of five postmenopausal women. One capsule of hops extract contained 0.25 mg of 8PN, 1.30 mg of 6PN, 0.80 mg of IX and 21.3 mg of XH. It was taken daily for 5 days (low dose). After one month, the dose was increased to two capsules daily for 5 days (medium dose) and after another month to four capsules daily for 5 days (high dose). The predominant derivatives of all compounds in serum were glucuronides with half-lives of at least 20 h. Increased serum and urine levels of IX compared with XN indicated its cyclization to IX in vivo. Bolca and colleagues [114] investigated the disposition of prenylflavonoids in breast tissues in women taking supplementation with hops extract (=0.1 mg 8 PN) for five days prior to aesthetic breast size reduction. XH, IX and 8PN were found to be present, mainly in the form of glucuronides.

Some human studies were done; however, their quality does not seem to always be at the required level (the lack of standardization of the extract, the recruitment of women from multiple sites, and the lack of a placebo effect). Hence, more quality clinical studies are needed to determine the effect of hops on menopausal symptoms.

### 2.6. Red Clover (Trifolium pratense L.)

Red clover or *Trifolium pretense* L. is a member of the Fabaceae family. It is native to Europe, Asia and Africa, and has been introduced to every other continent [115]. It is grown in terrestrial and wetlands, and is used for pasturage, hay and silage for the livestock [116]. It is a herbaceous, perennial plant, which grows up to 80 cm tall. The leaves are trifoliate (compounded typically of three leaflets) and alternate. The flowers are pink to red or white in color.

Red clover at the flowering stage contains isoflavones formononetin, biochanin A, daidzein and genistein in cumulative concentrations of 5.4–8.1 mg/g of dry matter, where formononetin and biochanin A contribute 51% and 40% of the weight, respectively [117]. Leaves contribute to 73.9% of the total isoflavone content, while stems contribute 17.6% and flowers approximately 9% [117]. In addition, glycitein and prunetin can also be found in red clover in smaller amounts [8]. Isoflavones act as phytoestrogens, as they activate ERs by binding to two isoforms: to estrogen receptor β (ERβ) with higher affinity and to estrogen receptor α (ERα) with lower affinity [8]. This may, in turn, lead to a reduction of gonadotropin-releasing hormone, follicle-stimulating hormone, and luteinizing hormone levels [118]. Additionally, isoflavones are thought to have antioxidant properties, to inhibit tyrosine kinases, and affect ion transport [8].

Breeding problems of sheep herds in Australia in 1940s, which fed on clover, brought attention to possible hormonal effects of this plant [119]. Many clinical studies were conducted to determine whether red clover could be used to help with menopausal symptoms due to its putative estrogenic activity. A recent (2021) systematic review and meta-analysis by Kanadys et al. presented randomized controlled trials on the use of red clover extract in menopause [8]. The effectiveness of red clover isoflavone extracts on the relief of hot flashes and menopausal symptoms in peri- and postmenopausal women was assessed. Eight trials out of 107 potentially relevant randomized controlled trials passed the quality criteria and were included in comparisons. In most of these trials, 40–80 mg of red clover isoflavone extract was given to participants per day. A meta-analysis revealed a reduction in the hot flashes frequency, by 1.73 hot flashes per day. The menopausal symptoms were alleviated upon treatment with red clover isoflavone extract according to Kupperman menopausal index and menopause rating scale, but not according to Greene climacteric scale. The red clover isoflavone extract was more effective in women experiencing more than five hot flashes per day, in doses higher than 80 mg of the extract per day, and when the content of biochanin A was higher.

Red clover is likely safe when used as a supplement to relieve menopausal symptoms [118]. No significant side effects were seen upon a year of use of such supplements. However, due to its estrogenic activity, patients on hormone replacement therapy or contraceptives, and patients with a history of hormone-dependent cancers should pay special attention to any adverse events [118]. Due to coumarin in red clover, it could have an effect on platelet aggregation, therefore special care is needed if used concomitantly with anticoagulants [118].

Overall, clinical studies support the use of a red clover isoflavone extract for women suffering from menopausal symptoms. Red clover supplements are considered safe for this indication.

### 2.7. Valerian (Valeriana officinalis L. s.l.)

*Valeriana* genus (Caprifoliaceae) is comprised of 289 species. The most important one, *Valeriana officinalis* L. (valerian), is known under at least 22 synonyms. It grows naturally in Europe and western Asia, and was introduced to North America. It prefers moist locations, but can also be found in drier soils. Morphology of valerian is very diverse. In the second year of growth, the plant produces a round, furrowed and hollow flowering stem, 80–120 cm tall and branched at the top. The pale green (upper side) lanceolate feathery leaves grow either from one feathered shape or from 9 to 21 finely serrated leaflets. Leaves are attached in pairs to either side of the stem. The stems terminate in umbels bearing many branches and tiny white and pale pink flowers. Valerian has a robust rhizome with many secondary roots and stolons.

The European Pharmacopoeia requires as a pharmaceutical material dried, whole or fragmented underground parts of valerian (*Valeriana officinalis* L. *s.l*.), including rhizome surrounded by the roots and stolons (*Valerianae radix*, [120]) or dried, cut underground parts of valerian, including rhizome, roots and stolons (*Valerianae radix minutata*, [121]). This use in folk medicine has been translated into official therapy, which is confirmed by the existence of a European herbal monograph [121]. Here, we can find indications “for the relief of mild nervous tension and sleep disorders“ in the well-established use category, and “for relief of mild symptoms of mental stress and to aid sleep“ in the traditional use category. The use of valerian root for nervous disorders during menopause is mentioned only by Usmanghani et al. (1997) [122].

Positive experiences with sedative and/or anxiolytic effects of valerian preparations, which could be employed during menopause, have aroused interest in finding the plant constituents responsible for these beneficial effects. Valerian root has been shown to contain several structurally distinct components that act by several different but complementary mechanisms of action on the central nervous system. Numerous in vitro experiments related to GABA receptors started in the 1990s and confirmed:

(a) Affinity of aqueous and hydroethanolic extracts with GABA_A_ receptors;

(b) Inhibition the uptake and stimulation the release of [^3^H]-GABA [123,124,125,126].

Recently, in silico inhibition of GABA aminotransferase (GABA-AT) by valerian compounds has been studied and compared with known GABA-AT inhibitors (vigabatrin and valproic acid). Isovaleric acid and didrovaltrate showed GABA-AT inhibitory activity, although they were less potent compared to vigabatrin [127].

A randomized, controlled, triple-blind study enrolled 100 women at least one year postmenopausal with insomnia and aged 50–60 years [128]. The capsules used in this study contained 530 mg of a valerian root extract. Participants took the capsules twice daily for four weeks. The mean sleep scale score before the intervention in the valerian group was 9.8 and after the intervention it was 6.02. The baseline score in the placebo group was 11.14 and post-placebo it was 9.4. Improvement in sleep quality was reported by approximately 30% of participants in the valerian group and 4% in the placebo group.

Another randomized double-blind clinical trial with 68 women aged 45 to 55 years with a chief complaint of hot flashes and taking 225 mg valerian capsules was described by Mirabiho et al. [129]. Data were collected pre-treatment, two weeks and four weeks post-treatment from two groups (valerian and placebo). Capsules were taken three times daily for eight weeks. At four and eight weeks after the treatment, the severity of hot flashes showed a significant difference between the two groups in favor of the valerian group.

Three years later, a one-month clinical trial was conducted on 129 menopausal women [130]. In each group (two experimental and one control group), there were 43 menopausal subjects who were not taking sedative drugs, did not smoke, and had no liver, kidney, or gastrointestinal disease, no history of cancer, and had not used hormone therapy within the last three months. The first experimental group used relaxation using the Benson method, the second used a commercial valerian product (530 mg), and the third group was a placebo. By comparing the sleep quality scores and their changes (before and after the intervention), the results showed an improvement in sleep disturbances only in the valerian and relaxation groups.

Next, a double-blind clinical trial was performed on women referring to a health center [131]. The sample size in this study included 48 women aged 45–62 years who were randomly divided into two groups, 29 in the valerian group and 19 in the placebo group. In the intervention group, women were treated with 350 mg capsules valerian every 12 h. The study lasted for two months and the control group with a placebo capsule similar to valerian was treated with the same treatment regimen. Information through demographic characteristics questionnaire, Hamilton anxiety scale and Beck depression inventory two months before treatment was completed. Based on the findings of the study, the two groups were not significantly different in terms of the severity of anxiety and depression symptoms at the beginning of the study, while after the intervention, the patients’ condition in terms of anxiety and depression in the valerian group was significantly lower.

The last known study was a triple-blind, randomized, controlled, two-month clinical trial with 60 female non-smokers, one to five years post-menopausal, aged 45–55 years [132]. Subjects took a 530 mg valerian capsule or a placebo twice daily for two months. The intensity of hot flashes in the valerian group was significantly lower than in the placebo group both one and two months after the start of the study.

All of the studies described here lack consistency. In addition, all of the above studies involved heterogeneous groups of women, some with chronic and acute insomnia, some with ongoing menopause, others post-menopausal. Therefore, further studies are needed to conclude whether valerian can help in alleviating central nervous system disorders in menopause.

### 2.8. Soybean (Glycine max (L.) Merr. and Glycine subsp. soja (Siebold & Zucc.) H. Ohashi)

Wild soybean or *Glycine soja* is an annual or perennial climbing herb from the legume family (Fabaceae) [133]. It is native in eastern Asia, Russian Far East, eastern China, Korean peninsula and Japan [134]. Wild soybean is not to be mistaken for widely cultivated domesticated soybean or *Glycine max*. In contrast to domesticated soybean, wild soybean has dormant seeds [135]. They are about 1.8–2.5 mm wide, 2.5–4.0 mm long, ellipsoid shape and black in color [136]. Domestic soybean has less genetic diversity and is more susceptible to damage from climatic changes, while wild soybean is more resilient due to its widespread presence in diverse climates and could consequently be a good resource for creating improved genetic variants of domesticated soybean [137].

Wild soybean beans contain a wide range of compounds, among them saponins and isoflavones (e.g., daidzein, 6-hydroxy-daidzein, daidzein glycosides, genistein, genistein glycosides, glycitein, and glycitein glycosides), trypsin inhibitors, and twice the amount of α-linolenic acid in triglycerides as domesticated soybean [135]. Glycoside forms of the three aglycons may be β-glucosides, 6″-*O*-malonyl-glucosides and 6″-*O*-acetyl-glucosides [138]. Isoflavonoid content in domesticated soybean per gram soybeans is up to: 516 µg daidzin, 1079 µg genistin, 177 µg glycitin, 768 µg malonyldaidzin, 158 µg malonylglycitin, 2446 µg malonylgenistin, 265 µg genistein [139]. A better absorption was shown for aglycone isoflavonoids than the glycoside forms in humans [140]. Soy isoflavones are of particular interest in the pharmaceutical industry. They mimic estrogen and are thus classified as phytoestrogens, and have antioxidant properties [140]. The estrogenic properties of some isoflavones are also the basis for the hypothesis that soy could act as a hormone replacement therapy and thus help alleviate menopausal symptoms. The typical isoflavone ingestion through supplements intended for the relief of menopausal symptoms is 35–150 mg/day [141]. Among soy isoflavones, genistein is the most potent with regard to ER binding, followed by daidzein [139,142]. Namely, genistein binds to ERβ with 30 times lower affinity than 17β-estradiol, and to ERα with a 10,000 times lower affinity [139]. This raises a question of whether soy isoflavones could act as selective ER modulators, i.e., exert estrogenic effects in some tissues, but none or antiestrogenic effects in other tissues [143].

The latter might be the reason that a vast body of literature, published on soy isoflavones, is very divergent and hard to interpret. Evidence suggesting a beneficial effect of soy food and soy extracts on menopausal symptoms and evidence opposing this claim have both been found. A systematic review and meta-analysis of randomized controlled trials from 2012 by Taku et al. [144] concluded that isoflavones at a median dose of 54 mg aglycone equivalents, taken for six weeks and up to a year, reduced the frequency of hot flashes by 20.6% and their severity by 26.6%. Supplements with larger doses of genistein (more than 18.8 mg) were found to be more effective in reducing the frequency of hot flashes than those containing lower doses of genistein [144]. Additionally, the authors attributed studies (either clinical trials or meta-analyses), which showed no improvement in hot flashes frequency, to the lack of differentiation of low and high genistein content in the studied supplements [144]. However, based on the EMA assessment report on domesticated soybean, finalized in 2018, clinical data on the use of ethanolic extracts of domesticated soybean are not sufficient to support the use of domesticated soybean for the relief of hot flashes and night sweating [139]. Most notable clinical studies on soy isoflavone extracts contained in medicinal products in the EU were done by Faure et al. and Stanosz et al. [139,145,146]. In the study by Faure et al., 75 menopausal women were given 70 mg standardized soy isoflavones per day for four months. A 61% reduction in hot flashes frequency was observed as compared with a 21% reduction in the placebo group. In the study by Stanosz et al., 71 women in early menopause were given two doses of an ethanolic soy extract (corresponding to 52 mg and 104 mg of genistein equivalents) for 12 months. The relief of symptoms occurred in the high dosage group after three months and in the low dosage group after five months, whereas the complete absence of hot flashes was reported in both, the high and low dosage groups after 12 months (compared to a 14% reduction in the placebo group). However, both studies had substantial shortcomings, e.g., high drop-out rate or no information on the drop-out rate, no information on time from the last period, poorly defined inclusion and exclusion criteria [139]. These shortcomings rendered these studies insufficient to confirm the efficacy of ethanol soy extracts [139].

Due to the hormone-like effects of isoflavones, they have been investigated for potential endocrine toxicity. However, a systemic review of over 400 studies, examining endocrine-related endpoints, by Messina et al. [119] showed that the evidence does not justify the criteria for isoflavones as endocrine disruptors. Risk assessment from the European Food Safety Authority concluded that isoflavones in typically ingested doses do not exert side effects on mammary gland, uterus and thyroid in peri- and postmenopausal women [141]. Similarly, the North American Menopause Society stated that isoflavones do not increase the risk for endometrial and breast cancer [143]. Caution should be exercised as soy is a known food allergen with a reported prevalence of 0.3–0.6% in adult population of the United States and Canada [147], and up to 0.5% in the general population aged 0–19 years [148]. A known interaction exists between soy food and levothyroxine, as soy food may decrease the absorption of levothyroxine. It is unknown if this interaction occurs with ethanolic extracts of soybean, as well [139].

Currently, the available evidence does not support the use of soy and derived products in the relief of menopausal symptoms. This is mostly due to the poor quality of the studies conducted. It appears, however, that genistein content plays a crucial role in the effectiveness of the soy-based supplement. As soy and derived products have a good safety profile (apart from being contraindicated in case of soy allergy or levothyroxine therapy), women suffering from hot flashes and night sweats may still try to alleviate them with soy supplements.

The drug part, active compounds, biological activities and supposed modes of action of the plants reviewed in this paper are summarized in Table 1.

## 3. Conclusions

Based on this review, we noted limited data are available on the use of some plants for alleviating the symptoms of menopause and gynecological disorders. While black cohosh and red clover were consistently shown to help reduce menopausal symptoms in clinical studies, currently available data do not fully support the use of fenugreek, hops, valerian, and soybean for this indication. For premenstrual syndrome and premenstrual dysphoric disorder, chaste tree shows effectiveness, but more clinical studies are needed to confirm such effect upon the use of evening primrose.

## Figures and Tables

**Table 1 molecules-26-07421-t001:** Commonly used plants in relieving menopausal symptoms.

Plant Species, Drug Part	Active Compounds	Biological Activities/Supposed Mechanism of Action
Black cohosh (*Cimicifuga racemosa)* rhizome	Phenolic compounds (ferulic acid, isoferulic acid and caffeic acid derivatives, cycloartane triterpene glycosides (actein, 26-deoxyactein, cimicifugoside)) and phenylpropanoids, possibly phytoestrogenic flavonoid formononetin, N_ω_-methylserotonin, 23-*O*-acetylshengmanol-3-*O*-d-xylopyranoside [15,19,20]	Modulation of key central nervous system receptors for thermoregulation, mood, and sleep (e.g., receptors for serotonin, dopamine, γ-aminobutyric acid (GABA), µ-opioids) [17,18] Improvement of metabolism in the brain and its overall activity [17,18] Modulating osteoclast growth and differentiation and mineralization [21,22]
Chaste tree (*Vitex agnus-castus*) fruit	Volatile compounds (essential oil), flavonoids and other phenolic compounds, iridoids, ketosteroids, chastol and epichastol diterpenoids [30,31,32,33] Methoxylated flavonol casticin, (also known as vitexicarpin) [34]	Binding to dopamine receptors followed by a decreased release of prolactin [36,37] Involvement of serotoninergic system has been proposed [38] Decreased serum prolactin levels [52]
Evening primrose (*Oenothera biennis*) seed	20% of oil (triglycerides) containing linoleic acid, γ-linolenic acid, palmitic acid, stearic acid, oleic acid, α-linolenic acid, unsaponifiable matter [34,58]	Modulation of the immune response and the synthesis of prostaglandins, cytokines and cytokine mediators [60]
Fenugreek (*Trigonella foenum-graecum*) seed	Polysaccharides (24–25% galactomannans), 0.016% essential oil, secondary metabolites (protoalkaloids, trigonelline, choline), 0.6–1.7% saponins (from diosgenin, yamogenin, tigogenin, and others), sterols (β-sitosterol), and flavonoids (orientin, isoorientin, isovitexin) [89]	Activation of the estrogen receptor (ER) [90] Upregulation of the expression of estrogen responsive genes [91] Proliferation of estrogen-dependent breast cells as well as antiproliferative effect on several cell lines [90,91,92,93] Increased plasma 17β-estradiol [90,94,98] Increased free testosterone and progesterone [98] Decreased in follicle stimulating hormone and steroid hormone binding globulin [98]
Hops (*Humulus lupulus*) inflorescence	Essential oil (constituents: β-myrcene, β-caryophyllene, α-humulene, β-farnesene, α-selinene, β-selinene, humulene epoxides, β-bisabolol, 2-methyl-3-buten-2-ol, a.s.o.), prenylated acylphloroglucinols (α-acids: humulone, its derivatives, and β-acids: lupulones), prenylated flavanones (isoxanthohumol, 6-prenylnaringenin, 8-prenylnaringenin), chalcones (xanthohumol, desmethylxanthohumol), triterpenes, flavonols, and tannins [100]	Estrogenic effect [105,106]
Red clover (*Trifolium pratense*) stem, leaf, flower	Isoflavones formononetin, biochanin A, daidzein and genistein, glycitein and prunetin [8,117]	Activation of the ERs by binding to two isoforms: to estrogen receptor β (ERβ) with higher affinity and to estrogen receptor α (ERα) with lower affinity [8] Reduction of gonadotropin releasing hormone, follicle stimulating hormone, and luteinizing hormone levels [118] Antioxidant activity, inhibition of tyrosine kinases and modulation of ion transport [8]
Valerian (*Valeriana officinalis*) rhizome, roots, stolons	Isovaleric acid and didrovaltrate [127]	Inhibition of GABA aminotransferase [127]
Soybean (*Glycine max* and *Glycine soja*) seed	Saponins and isoflavones (e.g., daidzein, 6-hydroxy-daidzein, daidzein glycosides, genistein, genistein glycosides, glycitein, and glycitein glycosides), trypsin inhibitors, and twice the amount of α-linolenic acid as domesticated soybean [135] Glycoside forms of the three aglycons may be β-glucosides, 6″-O-malonyl-glucosides and 6″-O-acetyl-glucosides [138]	Estrogenic effect–genistein binds to ERβ with 30 times lower affinity than 17β-estradiol, and to ERα with a 10,000 times lower affinity [139,140] Antioxidant activity [140]

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
