# Peer review of "Herbal Products Used in Menopause and for Gynecological Disorders"

_molecules, 2021, doi:10.3390/molecules26247421_

Round 1

Reviewer 1 Report

The authors report a complete review of the scientific literature on the use of herbal remedies to counteract the effects of pre-menopause and menopause. In particular, the authors report preclinical and clinical data on: Actearacemosa, Vitex agnus-castus, Oenothera biennis, Trigonella foenum-graecum, Humulus lupulus, Trifolium pratense, Valeriana officinalis, Glycine max.

The Review is interesting and clear, but needs some improvements:

1) The authors should better explain why they focused attention on these plant species;

2) It would be useful to insert a summary table showing the effects of the plant extracts observed and in which biological models;

3) It would be useful to insert or discuss better if the observed effects were attributable to the extraction methods used.

4) Finally, I suggest reporting the scientific name of the plant species the first time they appear (see abstract line 19-20)

5) In the introduction, line 28-29, the sentence requires a bibliographic reference.

Author Response

The authors report a complete review of the scientific literature on the use of herbal remedies to counteract the effects of pre-menopause and menopause. In particular, the authors report preclinical and clinical data on: Actearacemosa, Vitex agnus-castus, Oenothera biennis, Trigonella foenum-graecum, Humulus lupulus, Trifolium pratense, Valeriana officinalis, Glycine max.

The Review is interesting and clear, but needs some improvements:

1) The authors should better explain why they focused attention on these plant species;

We highly appreciate the reviewer's comment about the focus on the selected plants. We decided to present these plants after a comprehensive review of the literature. We included plants that are already in dietary supplements or OTC drug, which are advertised or have been shown to be effective in alleviating menopausal problems. We also included plants that have their own EMA (European Medicines Agency) monograph, which lists the effects on alleviating menopausal problems. We also included those plants for which we have found recent data in the literature that have not been reviewed in previous reviews and systematic articles. We have listed the sources and decision as to why we opted for specific plants at the end of the introduction chapter (lines 67-71)).

2) It would be useful to insert a summary table showing the effects of the plant extracts observed and in which biological models;

Thank you for this suggestion. A summary table (Table 1) has been added (line 954).

3) It would be useful to insert or discuss better if the observed effects were attributable to the extraction methods used.

Thank you for this comment. Generally, an extraction method (type of a method, type of a solvent, DER ratio in particular) significantly defines the qualitative and quantitative composition of an extract. To systematically evaluate the connection between the observed effects and extraction methods, the same herbal raw material is needed to perform different extraction variations. Since different herbal substances (used in the referenced studies) may greatly differ in their composition due to, for example, seasonal variations, geographical location, time of harvest, etc., together with the fact that in some studies herbal material was not phytochemically evaluated in detail, such discussion would be insufficient in our opinion. Therefore, we didn't include additional discussion in the manuscript.

4) Finally, I suggest reporting the scientific name of the plant species the first time they appear (see abstract line 19-20).

Thank you for this comment. Scientific names of the plant species were added to the abstract (lines 18-24).

5) In the introduction, line 28-29, the sentence requires a bibliographic reference.

Thank you for this remark. We have added an appropriate reference, which was lacking for this sentence (line 32):

Nedrow, A.; Miller, J.; Walker, M.; Ngyren, P.; Hoyt Huffman, L.; Nelson, H.D. Complementary and alternative therapies for the management of menopause-related symptoms: a systematic evidence review. Arch. intern. Med. 2006, 166(14), 1453-1465.

The authors report a complete review of the scientific literature on the use of herbal remedies to counteract the effects of pre-menopause and menopause. In particular, the authors report preclinical and clinical data on: Actearacemosa, Vitex agnus-castus, Oenothera biennis, Trigonella foenum-graecum, Humulus lupulus, Trifolium pratense, Valeriana officinalis, Glycine max.

The Review is interesting and clear, but needs some improvements:

1) The authors should better explain why they focused attention on these plant species;

We highly appreciate the reviewer's comment about the focus on the selected plants. We decided to present these plants after a comprehensive review of the literature. We included plants that are already in dietary supplements or OTC drug, which are advertised or have been shown to be effective in alleviating menopausal problems. We also included plants that have their own EMA (European Medicines Agency) monograph, which lists the effects on alleviating menopausal problems. We also included those plants for which we have found recent data in the literature that have not been reviewed in previous reviews and systematic articles. We have listed the sources and decision as to why we opted for specific plants at the end of the introduction chapter (lines 67-71)).

2) It would be useful to insert a summary table showing the effects of the plant extracts observed and in which biological models;

Thank you for this suggestion. A summary table (Table 1) has been added (line 954).

3) It would be useful to insert or discuss better if the observed effects were attributable to the extraction methods used.

Thank you for this comment. Generally, an extraction method (type of a method, type of a solvent, DER ratio in particular) significantly defines the qualitative and quantitative composition of an extract. To systematically evaluate the connection between the observed effects and extraction methods, the same herbal raw material is needed to perform different extraction variations. Since different herbal substances (used in the referenced studies) may greatly differ in their composition due to, for example, seasonal variations, geographical location, time of harvest, etc., together with the fact that in some studies herbal material was not phytochemically evaluated in detail, such discussion would be insufficient in our opinion. Therefore, we didn't include additional discussion in the manuscript.

4) Finally, I suggest reporting the scientific name of the plant species the first time they appear (see abstract line 19-20).

Thank you for this comment. Scientific names of the plant species were added to the abstract (lines 18-24).

5) In the introduction, line 28-29, the sentence requires a bibliographic reference.

Thank you for this remark. We have added an appropriate reference, which was lacking for this sentence (line 32):

Nedrow, A.; Miller, J.; Walker, M.; Ngyren, P.; Hoyt Huffman, L.; Nelson, H.D. Complementary and alternative therapies for the management of menopause-related symptoms: a systematic evidence review. Arch. intern. Med. 2006, 166(14), 1453-1465.

Reviewer 2 Report

The manuscript is well written but there are still some things that the authors need to include in order to improve its quality.  The materials and methods should be included. The databases where the information on the plants reported in this paper where obtained should be highlighted in the methods.  It would have been better to state those plants used in different region or countries around world used in the treatment of menopause disorders because these plants are not globally distributed.  The traditional ways of using these plants should be stated. If they are used orally, the toxicity effects on the liver, kidney, etc should be stated.  Are these plants used individually or in combination?  Table summarising some of the plants used for this disorder should be included. Let the paper contain some tables and figures.  The possible mechanisms of actions of the identified compounds in each of the reported plants should be stated or put in a schematic diagram.  The discussion section is not necessary; it should be integrated with the plants highlighted under results section. However, if the authors want to retain the manuscript is this format, the discussion is poorly written, it should be improved upon.  The toxicity effects of the reported plants should be reported because it is good to know the safety of the plants for human use.

Author Response

The manuscript is well written but there are still some things that the authors need to include in order to improve its quality.

  • The materials and methods should be included. The databases where the information on the plants reported in this paper where obtained should be highlighted in the methods.

We highly appreciate the reviewer's comment. Due to the fragmentation of the field we decided to adopt a horizontal review strategy. We included plants that are already in dietary supplements or OTC drug, which are advertised or have been shown to be effective in alleviating menopausal problems. We also included plants that have their own EMA (European Medicines Agency) monograph, which lists the effects on alleviating menopausal problems. We also included those plants for which we have found recent data in the literature that have not been reviewed in previous reviews and systematic articles. A review of literature on plants with activity on menopausal problems was conducted using PubMed, Google Scholar and plant regulatory authorities’ web pages. Publications in English, German and Slovenian were considered for screening. Since the journal does not require a chapter on Materials and Methods for review articles, we have not included it.

  • It would have been better to state those plants used in different region or countries around world used in the treatment of menopause disorders because these plants are not globally distributed.

We included plants that are already in dietary supplements or OTC drugs (present in the Western world) and are advertised or have been shown to be effective in alleviating menopausal problems. We also included plants that have their own EMA (European Medicines Agency) monograph, which lists the effects on alleviating menopausal problems.

  • The traditional ways of using these plants should be stated. If they are used orally, the toxicity effects on the liver, kidney, etc should be stated.

Because traditional plant use can be very undefined and present only in some parts of the world, only plants with contemporary use in the Western world (as dietary supplements or OTC drugs) were considered. For all the plants we described, we also listed side effects if they were described at the recommended dosage and route of administration, e.g., for fenugreek, where caution should be exercised in patients taking antidiabetic drugs, and which is contraindicated during pregnancy or red clover which could have an effect on platelet aggregation, therefore special care is needed if used concomitantly with anticoagulants.

  • Are these plants used individually or in combination?

These plants are often used in combinations, however this was out of the scope of this review, unless where the use of a combination was supported by a clinical study and contributed to better efficacy (e.g., as in the case of the combination of black cohosh and St. John's wort).

  • Table summarising some of the plants used for this disorder should be included. Let the paper contain some tables and figures.

Thank you for this suggestion. A summary table (Table 1) including the possible mechanisms of actions and active compounds has been added (lines 954).

  • The possible mechanisms of actions of the identified compounds in each of the reported plants should be stated or put in a schematic diagram.

Thank you for this suggestion. A summary table (Table 1) including the possible mechanisms of actions and active compounds has been added (lines 954).

  • The discussion section is not necessary; it should be integrated with the plants highlighted under results section. However, if the authors want to retain the manuscript is this format, the discussion is poorly written, it should be improved upon.

Thank you for this suggestion. The discussion was omitted, as these points were already included in each plant chapter.

  • The toxicity effects of the reported plants should be reported because it is good to know the safety of the plants for human use.

Safety of the reported plants was considered. For all the plants we described, we also listed side effects if they were described at the recommended dosage and route of administration, e.g., for fenugreek, where caution should be exercised in patients taking antidiabetic drugs, and which is contraindicated during pregnancy.

The manuscript is well written but there are still some things that the authors need to include in order to improve its quality.

  • The materials and methods should be included. The databases where the information on the plants reported in this paper where obtained should be highlighted in the methods.

We highly appreciate the reviewer's comment. Due to the fragmentation of the field we decided to adopt a horizontal review strategy. We included plants that are already in dietary supplements or OTC drug, which are advertised or have been shown to be effective in alleviating menopausal problems. We also included plants that have their own EMA (European Medicines Agency) monograph, which lists the effects on alleviating menopausal problems. We also included those plants for which we have found recent data in the literature that have not been reviewed in previous reviews and systematic articles. A review of literature on plants with activity on menopausal problems was conducted using PubMed, Google Scholar and plant regulatory authorities’ web pages. Publications in English, German and Slovenian were considered for screening. Since the journal does not require a chapter on Materials and Methods for review articles, we have not included it.

  • It would have been better to state those plants used in different region or countries around world used in the treatment of menopause disorders because these plants are not globally distributed.

We included plants that are already in dietary supplements or OTC drugs (present in the Western world) and are advertised or have been shown to be effective in alleviating menopausal problems. We also included plants that have their own EMA (European Medicines Agency) monograph, which lists the effects on alleviating menopausal problems.

  • The traditional ways of using these plants should be stated. If they are used orally, the toxicity effects on the liver, kidney, etc should be stated.

Because traditional plant use can be very undefined and present only in some parts of the world, only plants with contemporary use in the Western world (as dietary supplements or OTC drugs) were considered. For all the plants we described, we also listed side effects if they were described at the recommended dosage and route of administration, e.g., for fenugreek, where caution should be exercised in patients taking antidiabetic drugs, and which is contraindicated during pregnancy or red clover which could have an effect on platelet aggregation, therefore special care is needed if used concomitantly with anticoagulants.

  • Are these plants used individually or in combination?

These plants are often used in combinations, however this was out of the scope of this review, unless where the use of a combination was supported by a clinical study and contributed to better efficacy (e.g., as in the case of the combination of black cohosh and St. John's wort).

  • Table summarising some of the plants used for this disorder should be included. Let the paper contain some tables and figures.

Thank you for this suggestion. A summary table (Table 1) including the possible mechanisms of actions and active compounds has been added (lines 954).

  • The possible mechanisms of actions of the identified compounds in each of the reported plants should be stated or put in a schematic diagram.

Thank you for this suggestion. A summary table (Table 1) including the possible mechanisms of actions and active compounds has been added (lines 954).

  • The discussion section is not necessary; it should be integrated with the plants highlighted under results section. However, if the authors want to retain the manuscript is this format, the discussion is poorly written, it should be improved upon.

Thank you for this suggestion. The discussion was omitted, as these points were already included in each plant chapter.

  • The toxicity effects of the reported plants should be reported because it is good to know the safety of the plants for human use.

Safety of the reported plants was considered. For all the plants we described, we also listed side effects if they were described at the recommended dosage and route of administration, e.g., for fenugreek, where caution should be exercised in patients taking antidiabetic drugs, and which is contraindicated during pregnancy.

The manuscript is well written but there are still some things that the authors need to include in order to improve its quality.

  • The materials and methods should be included. The databases where the information on the plants reported in this paper where obtained should be highlighted in the methods.

We highly appreciate the reviewer's comment. Due to the fragmentation of the field we decided to adopt a horizontal review strategy. We included plants that are already in dietary supplements or OTC drug, which are advertised or have been shown to be effective in alleviating menopausal problems. We also included plants that have their own EMA (European Medicines Agency) monograph, which lists the effects on alleviating menopausal problems. We also included those plants for which we have found recent data in the literature that have not been reviewed in previous reviews and systematic articles. A review of literature on plants with activity on menopausal problems was conducted using PubMed, Google Scholar and plant regulatory authorities’ web pages. Publications in English, German and Slovenian were considered for screening. Since the journal does not require a chapter on Materials and Methods for review articles, we have not included it.

  • It would have been better to state those plants used in different region or countries around world used in the treatment of menopause disorders because these plants are not globally distributed.

We included plants that are already in dietary supplements or OTC drugs (present in the Western world) and are advertised or have been shown to be effective in alleviating menopausal problems. We also included plants that have their own EMA (European Medicines Agency) monograph, which lists the effects on alleviating menopausal problems.

  • The traditional ways of using these plants should be stated. If they are used orally, the toxicity effects on the liver, kidney, etc should be stated.

Because traditional plant use can be very undefined and present only in some parts of the world, only plants with contemporary use in the Western world (as dietary supplements or OTC drugs) were considered. For all the plants we described, we also listed side effects if they were described at the recommended dosage and route of administration, e.g., for fenugreek, where caution should be exercised in patients taking antidiabetic drugs, and which is contraindicated during pregnancy or red clover which could have an effect on platelet aggregation, therefore special care is needed if used concomitantly with anticoagulants.

  • Are these plants used individually or in combination?

These plants are often used in combinations, however this was out of the scope of this review, unless where the use of a combination was supported by a clinical study and contributed to better efficacy (e.g., as in the case of the combination of black cohosh and St. John's wort).

  • Table summarising some of the plants used for this disorder should be included. Let the paper contain some tables and figures.

Thank you for this suggestion. A summary table (Table 1) including the possible mechanisms of actions and active compounds has been added (lines 954).

  • The possible mechanisms of actions of the identified compounds in each of the reported plants should be stated or put in a schematic diagram.

Thank you for this suggestion. A summary table (Table 1) including the possible mechanisms of actions and active compounds has been added (lines 954).

  • The discussion section is not necessary; it should be integrated with the plants highlighted under results section. However, if the authors want to retain the manuscript is this format, the discussion is poorly written, it should be improved upon.

Thank you for this suggestion. The discussion was omitted, as these points were already included in each plant chapter.

  • The toxicity effects of the reported plants should be reported because it is good to know the safety of the plants for human use.

Safety of the reported plants was considered. For all the plants we described, we also listed side effects if they were described at the recommended dosage and route of administration, e.g., for fenugreek, where caution should be exercised in patients taking antidiabetic drugs, and which is contraindicated during pregnancy.